# Structure of the hyperosmolality-gated calcium-permeable channel OSCA1.2

Xin Liu[1], Jiawei Wang[2] & Linfeng Sun[1,3]

In plants, hyperosmolality stimuli triggers opening of the osmosensitive channels, leading to a rapid downstream signaling cascade initiated by cytosolic calcium concentration elevation. Members of the OSCA family in *Arabidopsis thaliana*, identified as the hyperosmolality-gated calcium-permeable channels, have been suggested to play a key role during the initial phase of hyperosmotic stress response. Here, we report the atomic structure of *Arabidopsis* OSCA1.2 determined by single-particle cryo-electron microscopy. It contains 11 transmembrane helices and forms a homodimer. It is in an inactivated state, and the pore-lining residues are clearly identified. Its cytosolic domain contains a RNA recognition motif and two unique long helices. The linker between these two helices forms an anchor in the lipid bilayer and may be essential to osmosensing. The structure of AtOSCA1.2 serves as a platform for the study of the mechanism underlying osmotic stress responses and mechanosensing.

---

[1] Hefei National Laboratory for Physical Sciences at Microscale, School of Life Sciences, University of Science and Technology of China, 230027 Hefei, China. [2] State Key Laboratory of Membrane Biology, Beijing Advanced Innovation Centre for Structural Biology, School of Life Sciences, Tsinghua University, 100084 Beijing, China. [3] CAS Centre for Excellence in Molecular Cell Science, University of Science and Technology of China, Chinese Academy of Sciences, 230027 Hefei, China. Correspondence and requests for materials should be addressed to J.W. (email: jwwang@mail.tsinghua.edu.cn) or to L.S. (email: sunlf17@ustc.edu.cn)

Osmotic stress is an important environmental factor that affects all living organisms. Cells exhibit a wide range of sensors and signalling networks at different levels to adjust to extreme conditions when exposed to either hyperosmotic or hypoosmotic environments[1–4]. In bacteria, several channels, like the structurally elucidated mechanosensitive channels of large and small conductance (MscL and MscS, respectively) proteins, work as osmotic safety valves and tune the cell responses to osmotic shock[5,6]. In mammals, the mechanosensitive channels that may play roles in osmosensing include the potassium channels TREK-1 and TRAAK, TRP family channels like TRPV4 and TRPC6, and the well-known Piezo channels[7–10]. During channel activation, the membrane tension induced by osmotic stress is converted to ion flux and osmolytes release. In plants and mammals, calcium ion acts as a primary regulator of the initial responses to osmotic pressure[11–16]. The first event observed after osmotic stress treatment is a rapid increase in the cytosolic free $Ca^{2+}$ concentration[17,18]. The molecular identities of osmosensors in plants include the MscS-like proteins and MCA family proteins[11,19–23]. Via genetic screens and functional analysis, the osmosensitive $Ca^{2+}$ permeable cation channel proteins were identified in Arabidopsis by two independent groups, which was named CSC1 (alias OSCA1.2 according to the OSCA family nomenclature that is used here) and OSCA1, respectively[17,24]. The OSCA family in Arabidopsis consists of 15 protein members with sequence identities varying from 14% to 85% (Supplementary Fig. 1a). OSCA1.2 and OSCA1 share ~85% sequence identity, and they both can be activated by hyperosmolality treatment. In addition to $Ca^{2+}$, they are also permeable to monovalent cations such as $Na^+$ and $K^+$, suggesting low cation selectivity[17,24]. A genome-wide survey of the essential crop Oryza sativa (Asian rice) also identified 11 genes encoding OSCA1 homologues with tissue-specific expression profiles[25]. Some genes are specifically expressed in stomatal guard cells and transcriptionally regulated by the circadian clock, suggesting the tight regulation of water potential during the day and night[25]. In addition to plants, homologues have also been identified in yeast and human[24,26]. TMEM63A, TMEM63B and TMEM63C are three orthologues in human that may function in osmoreception.

Members of the OSCA family belong to the calcium-permeable stress-gated cation channel family (entry number of 1.A.17.5 in Transporter Classification Database), a subfamily of the calcium-dependent chloride channel (Ca-ClC) family[27]. Representatives of the Ca-ClC family include calcium-activated chloride channels such as TMEM16A and transmembrane channel-like proteins such as Tmc1. Previous structural studies of Ca-ClC family proteins, such as the lipid scramblase nhTMEM16 and the anion channel mTMEM16A, reveal that the protein forms a homodimer, and each monomer contains 10 transmembrane segments (TMS)[28–32]. Two ion conduction pores are present in the dimeric mTMEM16A and function independently in terms of channel activation and ion conduction[33]. Since OSCA and TMEM16A have discrete biological functions and share low sequence similarity, whether they have similar structures remains to be elucidated. The molecular mechanisms for hyperosmolality sensing and $Ca^{2+}$ permeation mediated by OSCA also remain largely unknown.

Here, we present the structure of OSCA1.2 from Arabidopsis thaliana (AtOSCA1.2) determined by single-particle cryo-electron microscopy (cryo-EM) with a nominal resolution of 3.68 Å for the overall structure. This structure serves as a framework for understanding OSCA-mediated osmosensing and calcium flux in plants at a molecular level.

## Results

**Structure determination of AtOSCA1.2.** We tried to express the full-length OSCA1.2 of A. thaliana, which contains 771 amino acids, using a baculovirus-based insect cell expression system and purify the protein. The protein displayed good solution behaviour as determined by gel filtration and EM imaging (Supplementary Fig. 2a, b). We then proceeded to determine its structure by single-particle cryo-EM analysis (Supplementary Fig. 2c, d and 3a). Eventually, we obtained an EM map with an overall resolution of 3.68 Å, according to the gold-standard Fourier shell correlation (FSC) criterion (Fig. 1a and Supplementary Fig. 2f). Local resolution analysis revealed that most of the transmembrane region and cytosolic domain had a resolution within 3.7 Å, and a small portion of the cytosolic domain and extracellular loops had a lower resolution (Supplementary Fig. 2e). The high quality of the EM density allowed us to carry out de novo model building (Supplementary Figs. 2g and 3b). In total, 683 residues were structurally modelled, and 670 side chains were assigned reliably. The 54 carboxyl-terminal residues are missing in the final structure, probably due to their structural flexibility.

**Overall structure of AtOSCA1.2.** AtOSCA1.2 forms a homodimer, similar to nhTMEM16 and mTMEM16A. The overall structure is ~85 Å in height and ~140 Å in width (Fig. 1a, b). Each monomer contains 11 transmembrane helices, in contrast to nhTMEM16 and mTMEM16A, which have only 10 TMS (Fig. 2a, b). The extracellular part of the protein is relatively small, mainly composed of loops connecting adjacent TMS. The cytosolic domain of AtOSCA1.2 is larger and mainly formed by two components: the linker between TM3 and TM4, containing approximately 175 amino acids, and the carboxyl-terminus after TM11 (Fig. 2b). In contrast to nhTMEM16 and mTMEM16A, of which the dimer formation is mediated by direct interaction between the transmembrane helices TM3 and TM10, the dimer interface of AtOSCA1.2 is exclusively formed by its cytosolic domain. Interestingly, when the dimeric AtOSCA1.2 structure is compared with that of nhTMEM16 or mTMEM16A, one subunit of AtOSCA1.2 aligns quite well, whereas the other subunit exhibits a large shift with both rotation and translation movements (Supplementary Fig. 4a). The transmembrane regions and extracellular parts of the monomers are not involved in dimer formation and are separated with a closest distance of ~8 Å (Supplementary Fig. 4b). Consequently, a large cavity is formed between the two monomers, which opens to the extracellular side and is sealed from the cytoplasm (Fig. 2c). In nhTMEM16 and mTMEM16A, a similar cavity, named the dimer cavity, is also observed at the dimer interface. However, it is divided into two furrows at the extracellular side.

The dimer cavity of AtOSCA1.2 has a diameter ranging from 8 to 20 Å. The surface of the cavity is mainly hydrophobic in nature. The inner part of the cavity on the cytoplasmic side contains several positively charged residues that may interact with the negatively charged head groups of certain lipids (Fig. 2c). Within the transmembrane region, the dimer cavity is fully accessible to the membrane through two V-shaped gates formed by TM4 of one subunit and TM11 of another (Fig. 2c and Supplementary Fig. 4b). At a low threshold, electron density can be observed within the cavity, likely attributable to lipid or detergent molecules. As proposed previously, the dimer cavity of nhTMEM16 or mTMEM16A is unlikely to be the catalytic site for lipid scrambling or the anion conduction pore[28,32]. Whether the dimer cavity has any role in the protein function remains unclear.

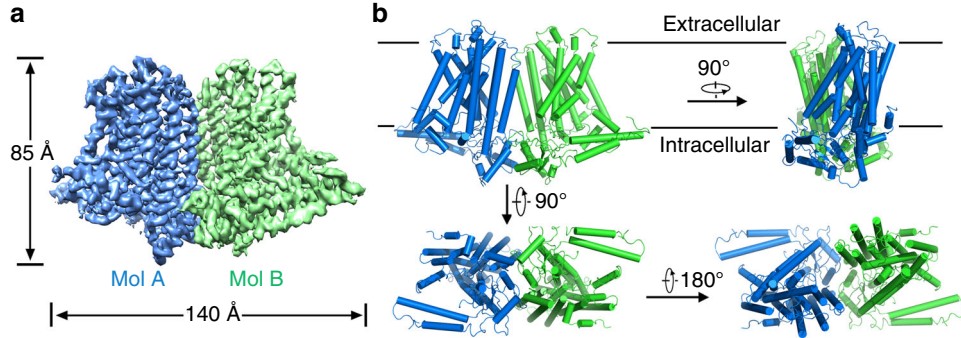

**Fig. 1** Overall dimeric structure of AtOSCA1.2. **a** An overview of the EM density at 3.68 Å resolution. Densities for the two AtOSCA1.2 molecules in the dimer structure are coloured blue and green, respectively. **b** The cartoon representation of the AtOSCA1.2 structure is shown in four perpendicular views. The two AtOSCA1.2 monomers are coloured blue and green, respectively. Each monomer contains 11 transmembrane helices. All structure figures were prepared using PyMol[66]

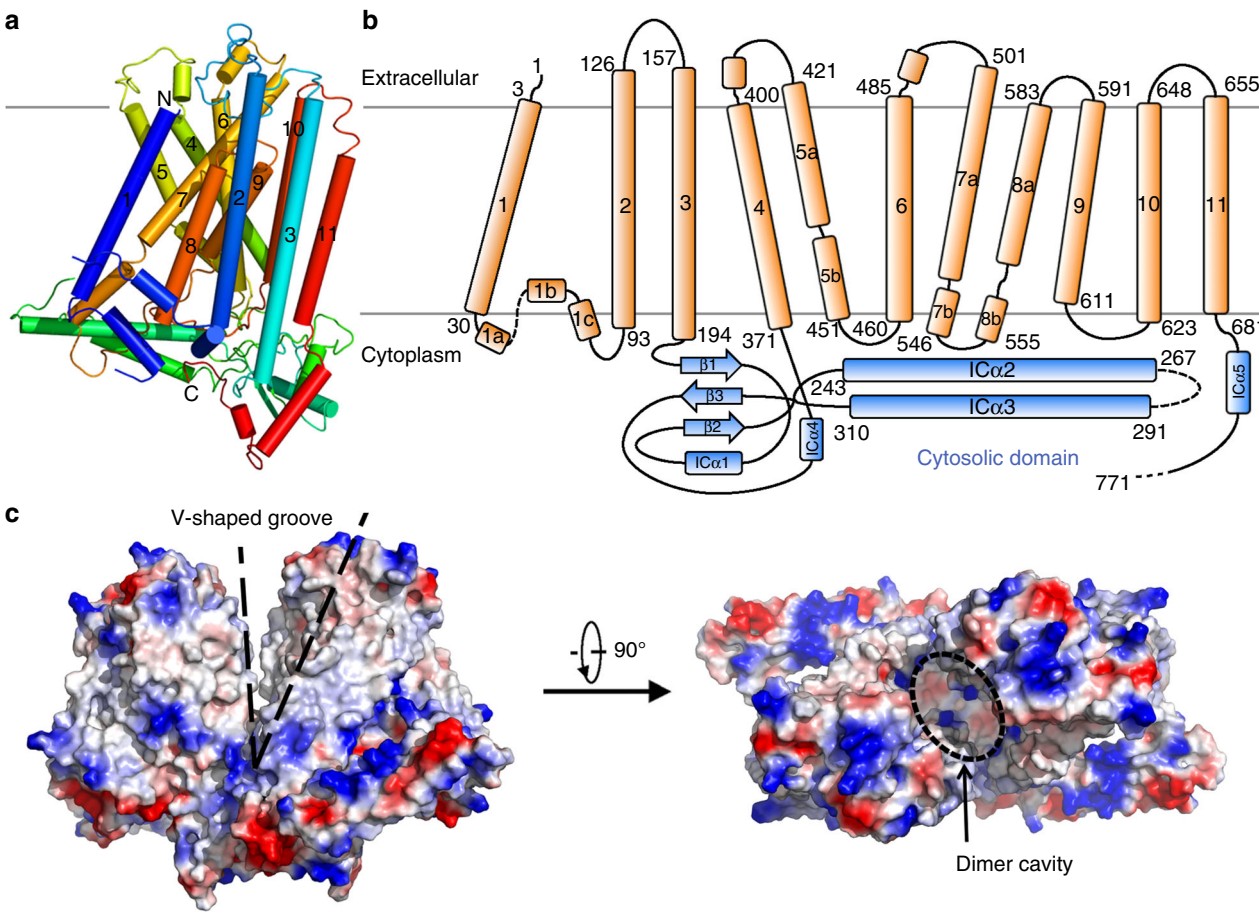

**Fig. 2** Structural features of AtOSCA1.2. **a** The overall structure of AtOSCA1.2 monomer. The 11 TMS are rainbow coloured, with the amino-terminus in blue and the carboxyl-terminus in red. **b** Topological diagram of AtOSCA1.2. The transmembrane region is coloured orange and the cytosolic domain is coloured blue. **c** A side view of the V-shaped groove and an extracellular view of the dimer cavity. The surface electrostatic potential was calculated with PyMol

**Transmembrane region of AtOSCA1.2.** Despite the separate biological functions and low sequence similarity between the AtOSCA1.2 and TMEM16 proteins (~11% sequence identity between AtOSCA1.2 and mTMEM16A), structural alignments reveal that the last 10 TMS of AtOSCA1.2 share a similar fold with nhTMEM16 and mTMEM16A (Fig. 3b). The pore-forming helices contain TM4–TM8, as inferred from mTMEM16A. The extra TM, TM1 in AtOSCA1.2, lies parallel to TM7, right beside the conduction pore (Fig. 3a). Its amino-terminus is in close

contact with TM5 and TM7, which further narrows the pore entrance on the extracellular side. Following TM1, there is a short helical structure that forms hydrophobic and polar interactions with the intracellular parts of TM7, TM8 and TM9. The linker between TM1 and TM2 contains 63 amino acids, 22 of which are missing in the structure model. Intriguingly, two short helices prior to TM1 of nhTMEM16 and mTMEM16A (corresponding to TM2 of AtOSCA1.2) are also present in AtOSCA1.2. They form a hairpin structure in nhTMEM16 and mTMEM16A but

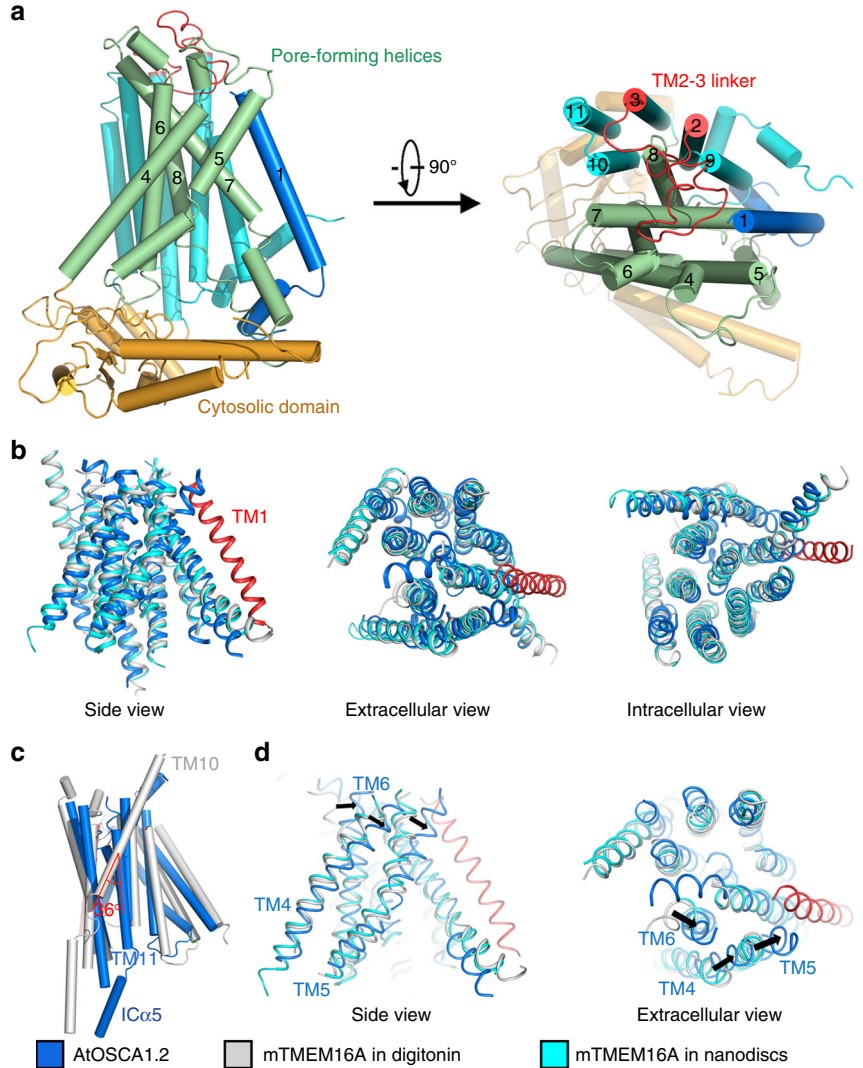

**Fig. 3** Structure comparison between AtOSCA1.2 and mTMEM16A. **a** Structural elements of AtOSCA1.2. The first TM of AtOSCA1.2 is coloured blue. The pore-forming helices of AtOSCA1.2 contain TM4–TM8 and is coloured green. TM9–TM11 are coloured cyan. The cytosolic domain is coloured orange. A long linker between TM2 and TM3 is coloured red. The structure is presented in a side view (left) and an extracellular view (right). **b** Structural alignments of the transmembrane domain of AtOSCA1.2 and mTMEM16A. In contrast to mTMEM16A, AtOSCA1.2 contains an extra TM, TM1, which is coloured red. The structure of AtOSCA1.2 resembles the Ca$^{2+}$-bound mTMEM16A structure determined in digitonin (coloured grey, PDB code: 5OYB (https://doi.org/10.2210/pdb5OYB/pdb)) or nanodiscs (coloured cyan, PDB code: 6BGI (https://doi.org/10.2210/pdb6BGI/pdb)). Three views are presented: a side view (left), an extracellular view (middle) and an intracellular view (right). **c** A 36° rotation is observed for the TM11 of AtOSCA1.2 comparing with TM10 of mTMEM16A. AtOSCA1.2 and mTMEM16A are coloured blue and grey, respectively. **d** The relative movement of TM4–TM6 in AtOSCA1.2. TM4–TM6 in AtOSCA1.2 move counterclockwise as viewed from the extracellular side, comparing with mTMEM16A in digitonin (PDB code: 5OYB (https://doi.org/10.2210/pdb5OYB/pdb)) or nanodiscs (PDB code: 6BGI (https://doi.org/10.2210/pdb6BGI/pdb)). The relative movement leads to further closure of the pore on the extracellular side. TM1 of AtOSCA1.2 is coloured red

are pulled apart and almost perpendicular to each other in AtOSCA1.2 (Supplementary Fig. 5).

Despite sharing the same fold, AtOSCA1.2 and mTMEM16A still exhibit dramatic differences. The most obvious one lies in the last TM. TM11 of AtOSCA1.2 is rotated approximately 36° around a pivot point in the middle of the helix, compared with TM10 of mTMEM16A (Fig. 3c). In mTMEM16A, the extracellular parts of TM10 from the two subunits interact with each other to form the dimer interface. Owing to the rotation, TM11 of AtOSCA1.2 no longer interacts with its counterpart, leading to the difference in dimer formation as discussed above. Another difference lies in the pore-forming helices, TM4–TM6 of

AtOSCA1.2 and TM3–TM5 of mTMEM16A. In mTMEM16A, the extracellular parts of TM3 and TM4 are shifted in the structure determined using lauryl maltose neopentyl glycol compared with the structures determined using digitonin and nanodiscs, resulting in a narrower pore. In AtOSCA1.2, the corresponding helices, TM4 and TM5, are further shifted, and a similar displacement is observed for TM6 (Fig. 3d). The relative movement of TM4–TM6 in AtOSCA1.2 leads to closure of the pore on the extracellular side, resulting in a much narrower pore than mTMEM16A (Fig. 3d). These three helices tend to be more mobile and may be linked to the conformational changes to gate the channel.

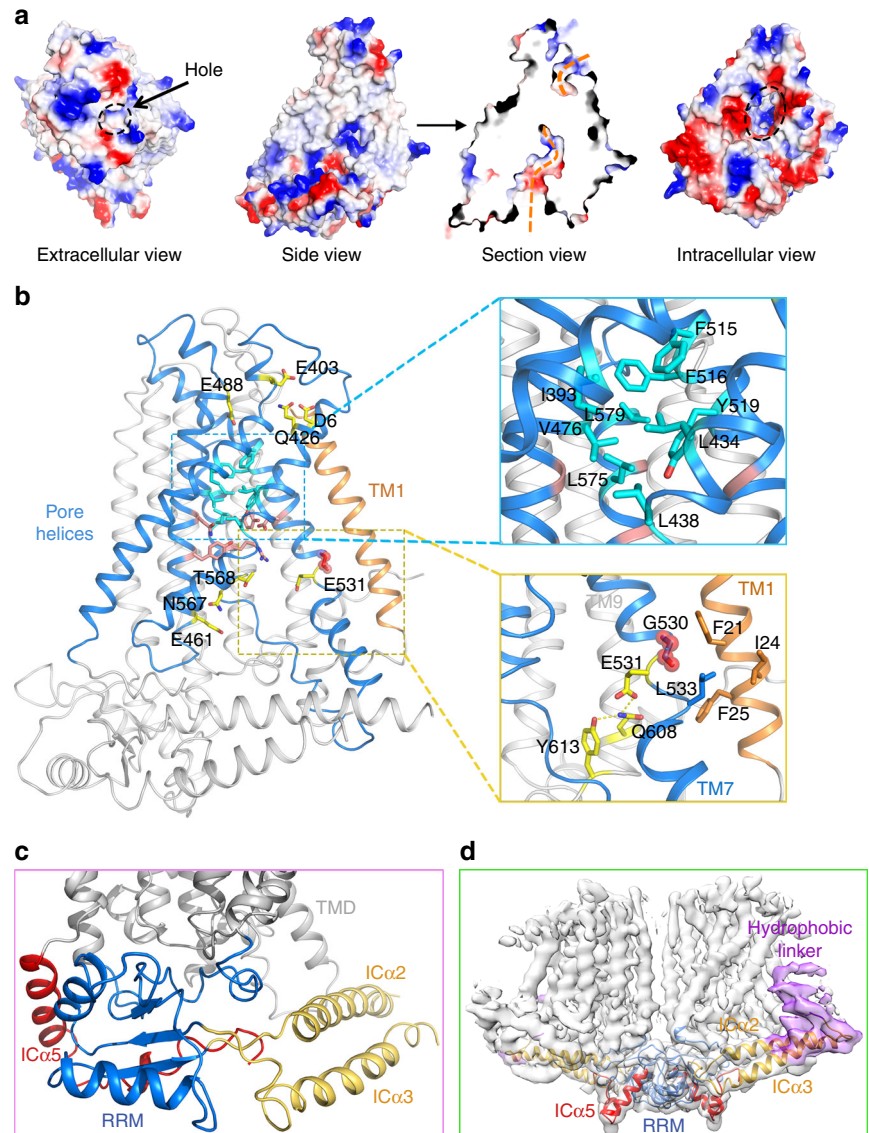

**Fig. 4** The ion conduction pore of AtOSCA1.2. **a** Surface view of the ion conduction pore of AtOSCA1.2. The extracellular part of the pore is relatively small as indicated by a black dashed-line circle. The conduction pore is in a closed state and blocked in the middle as shown in the section view. It is accessible from the cytoplasmic side through a hole as indicated in the intracellular view. **b** Pore-lining residues of AtOSCA1.2. The extracellular part of the pore contain several negatively charged or polar residues, as shown in sticks coloured yellow. The pore is constricted by hydrophobic residues in the middle (coloured cyan), as shown in a zoom-in view. The hinge region on TM7 is also shown in a zoom-in view. The hinge residue, G530, is shown in spheres. E531 of AtOSCA1.2 forms a triad with Q608 and Y613 via hydrogen bonds. The inner part of the conduction pore contains several negatively charged or polar residues, like E461, N567 and T568, as shown in sticks. **c** The cytosolic domain of AtOSCA1.2 contains an RNA recognition motif (RRM, coloured blue), a short helix after TM11 (ICα5, coloured red), and two long helices that lie parallel to the membrane plane (ICα2 and ICα3, coloured yellow). **d** The electron density for the linker region between ICα2 and ICα3 in a 5 Å low-pass filtered map (coloured purple). This region shows a high degree of flexibility and protrudes into the membrane like an anchor

**Ion conduction pore of AtOSCA1.2.** The ion conduction pore of AtOSCA1.2 is in a closed state in the solved structure, as revealed by pore radius analysis (Fig. 4a and Supplementary Fig. 6a, b). On the extracellular side, it is partly blocked by a long linker between TM2 and TM3, which forms strong interactions with the pore-forming helices (Fig. 3a). The entrance of the pore is formed by mainly negatively charged or polar residues, such as D6 on TM1, E403 on TM4, Q426 on TM5 and E488 on TM6, which may contribute to ion selection by lowering the energy barrier for cation passage (Fig. 4b). The conduction pore is constricted by a set of hydrophobic residues, which include I393 on TM4, L434 and L438 on TM5, V476 on TM6, F515, F516 and Y519 on TM7 and L575 and L579 on

TM8 (Fig. 4b). Such hydrophobic residues are also identified in mTMEM16A at similar positions and can affect the anion selectivity or channel gating, as determined by electrophysiological analysis[31]. Right beneath the hydrophobic-constricting region, the pore consists of several polar or negatively charged residues, like Y468 and N472 on TM6, Y519 and D523 on TM7 and Y576 on TM8 and a positively charged residue R572 on TM8 (Supplementary Fig. 7). D523 interacts with Y519 and Y576 via hydrogen bonds, respectively. R572 forms a hydrogen bond with Y605 on TM9. Y519 and D523 point towards the pore and may be directly involved in cation recognition and passage. Y576 helps to fix them in such a conformation through hydrogen bond formation.

The inner part of the pore is unsealed to the membrane environment due to the separation of TM5 and TM7. Accordingly, a groove extends nearly halfway into the lipid bilayer, smaller than that observed in nhTMEM16 but much larger than that in mTMEM16A (Supplementary Fig. 6c). Structural comparison of mTMEM16A and AtOSCA1.2 shows that the difference in groove size is mainly caused by a difference in TM7 of AtOSCA1.2 (corresponding to TM6 of mTMEM16A). mTMEM16A contains a hinge region on TM6 around residue G644[30]. $Ca^{2+}$ binding induces a dramatic conformational change in TM6. Notably, in AtOSCA1.2, TM7 is also non-continuous and broken after residue G530 (Fig. 4b). The region after G530 shifts laterally and moves farther away from TM5.

In mTMEM16A, calcium binding induces a large conformational change in TM6 (corresponding to TM7 of AtOSCA1.2) and activates the channel[30]. In our structure, TM7 of AtOSCA1.2 resembles the $Ca^{2+}$-bound, activated conformation observed in mTMEM16A[30] (Supplementary Fig. 6d). However, according to sequence and structure analysis, AtOSCA1.2 lacks the $Ca^{2+}$-binding sites. The observed state of TM7 of AtOSCA1.2 in the absence of $Ca^{2+}$ may be due to its strong interactions with surrounding helices, such as TM1, TM8 and TM9. In particular, its carboxyl-terminus also interacts with a cytosolic helix (Supplementary Fig. 6e). A conserved glutamic acid residue is found near the hinge region, namely, E531 in AtOSCA1.2, corresponding to E654 of mTMEM16A. Instead of forming a $Ca^{2+}$-binding site as in mTMEM16A, E531 of AtOSCA1.2 forms a triad with Q608 and Y613 via hydrogen bonds (Fig. 4b). The inner part of the conduction pore contains several negatively charged or polar residues, such as E461 and N472 on TM6, D523 on TM7 and N567 and T568 on TM8 (Fig. 4b). The ion conduction pore is fully accessible from the intracellular side through a hole formed by the cytosolic domain and transmembrane helices (Fig. 4a). The diameter of this hole is >10 Å, and the surface potential is mostly negative, which is favourable for cation passage.

**Cytosolic domain of AtOSCA1.2.** An extensive search of the Protein Data Bank (PDB) for similar structures by the DALI server led to the identification of an RNA recognition motif (RRM) in the cytosolic domain of AtOSCA1.2[34]. This RRM contains three anti-parallel β-sheets and two α-helices (Fig. 4c and Supplementary Fig. 8a). Previous computational analysis of an early responsive to dehydration protein (ERD4), which shares 27% sequence identity with AtOSCA1.2, suggested the presence of a sequence that can form two RRM motifs[35]. This sequence is conserved in AtOSCA1.2. However, only one RRM is identified in our structure, while the remaining part of this sequence forms two long helices. The ability of AtOSCA1.2 to bind RNA remains uncertain. However, the classical β-sheet surface for RNA binding is occupied in the current state by the loop prior to TM4 and the carboxyl-terminus of AtOSCA1.2 (Supplementary Fig. 8b). Unless a conformational change occurs in this region to expose the binding surface, AtOSCA1.2 is unlikely to bind RNA. As seen in the structure, two α-helices of the RRM (ICα1 and ICα4) together with the short helix following TM11 (ICα5) play mainly a structural role in mediating the dimer formation of AtOSCA1.2 (Supplementary Fig. 4c). Intense hydrophobic interactions are formed by W331, V335 and L685. A pair of hydrogen bond is also identified between Q340 and R682 (Supplementary Fig. 4d).

We have also identified two long helices in the cytosolic domain of AtOSCA1.2 that lie parallel to the membrane plane (named ICα2 and ICα3). As observed in the cryo-EM structure, they are partly buried in the micelle and may form direct contacts with the detergent molecules. This region contains several

positively charged residues, including five consecutive lysine residues, which may interact with the negatively charged head groups of lipids. On one end of the two helices, they are adjacent to TM4 and TM5 and tightly packed together with the RRM. On the other, ICα2 also interacts with the cytosolic part of TM7 (Supplementary Fig. 6e). The linker between ICα2 and ICα3 exhibits relatively poor EM density because of its intrinsic flexibility. We were not able to assign the side chains for the linker between ICα2 and ICα3, and 12 amino acids are missing in the atomic model. In the low-pass-filtered map, the linker can be observed mostly buried in the lipid bilayer like an anchor (Fig. 4d). It draws our attention as it represents a distinguishable structural element identified in Ca-ClC family proteins and mechanosensitive channels that may serve as a sensor of membrane stress. It contains a hydrophobic sequence, LGFLGLWG (278–285), which is highly conserved in most OSCA family proteins (Supplementary Fig. 1b). As shown by sequence alignment, the hydrophobic linker is highly conserved in AtOSCA1.1-1.8, AtOSCA2.3 and AtOSCA3.1, with an GXXGXXG motif (X typically represents a hydrophobic residue), except in AtOSCA2.1-2.2, AtOSCA2.4-2.5 or AtOSCA4.1. We sought to determine a structure of these exceptions and were able to obtain AtOSCA2.2 protein with a reasonable yield and good behaviour. By single-particle cryo-EM analysis, the structure of

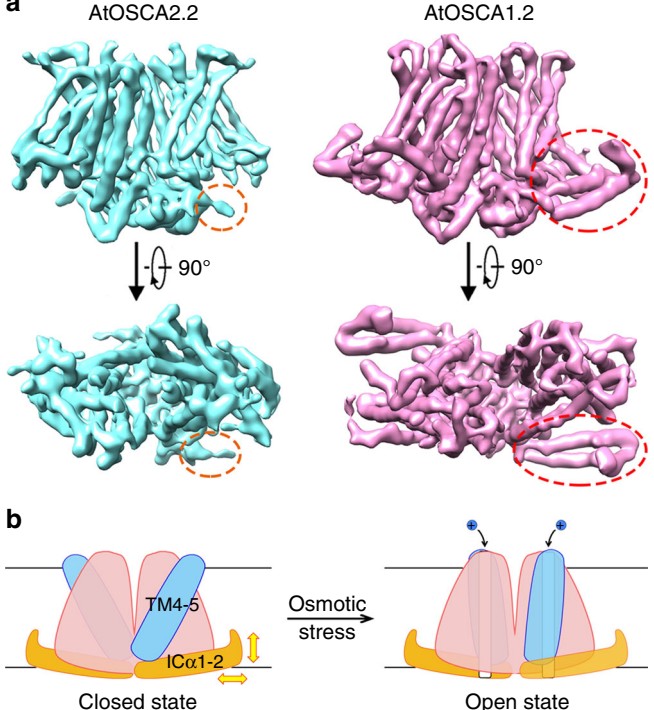

**Fig. 5** Comparison of the EM maps between AtOSCA1.2 and AtOSCA2.2. **a** The two long helices and the anchor region in the cytosolic domain are missing in the structure of AtOSCA2.2. A structure of AtOSCA2.2 was determined at 5.4 Å resolution by single-particle cryo-EM analysis. Comparing with AtOSCA1.2, AtOSCA2.2 lacks the two long helices and the hydrophobic linker in its cytosolic domain, as indicated by the dashed-line circles. A 5.4 Å low-pass filtered map for AtOSCA1.2 is presented here for comparison. The structures of AtOSCA2.2 and AtOSCA1.2 are coloured cyan and magenta, respectively. **b** A cartoon model for AtOSCA1.2 activation upon osmotic stress. The hydrophobic linker between the two long cytosolic helices ICα1-2 is proposed to serve as a sensor of osmotic stress. Its local motions triggered by the membrane tension and distortion upon osmotic stimulus may be linked to the tilting and rearrangements of the pore-forming helices such as TM4-5, resulting in channel opening

AtOSCA2.2 was determined with an overall resolution of 5.4 Å. The EM density for the transmembrane region of AtOSCA2.2 fits well with AtOSCA1.2. However, the two long helices and the anchor region in the cytosolic domain seem to be missing in AtOSCA2.2 (Fig. 5a). Whether this structural element functions to sense the membrane tension and different OSCA proteins adopt distinct activation mechanisms awaits further characterization.

## Discussion

In this study, we determined the structure of the hyperosmolality-gated calcium-permeable channel OSCA1.2 from *A. thaliana* at an atomic resolution. When exposed to stress conditions such as drought or a high-salinity environment, plant cells can respond within seconds, as indicated by an increase in cytosolic $Ca^{2+}$, followed by abscisic acid accumulation and other downstream signalling pathways[2,36]. As a subtype of mechanical forces, osmotic stress generates membrane tension, which applies forces to the embedded channels. Multiple models have been proposed for the gating mechanism of mechanosensitive channels, such as the lipid-disordering model, in which conformational changes of the channel help to reduce the local lipid deformation[37], and the hydrophobic mismatch model, in which conformational changes occur to accommodate the hydrophobic environment mismatch due to membrane thinning[38,39]. A certain element of the protein senses the membrane tension and triggers conformational changes to adapt to these forces[40,41]. In the bacterial MscL and MscS channels, amphipathic structural elements, such as the N-helix and β-hairpin structure in MaMscL and TM3b in EcMscS, have been proposed to function as membrane tension sensors and to play an anchoring role for the TMS[42–46]. Local motion of the sensor results in conformational changes in the pore region and regulates channel activation.

In AtOSCA1.2, the hydrophobic linker between the two long cytosolic helices is identified as an unique structural component and may serve as a sensor of osmotic stress. In the structure, this linker protrudes into the lipid bilayer as an anchor, and its partial deletion impairs the opening of the channel under hyperosmotic stress. Local motions of the linker and the two cytosolic helices caused by the membrane tension and distortion may be connected with rearrangements of the pore-forming helices such as TM7 and TM4–TM5, which form direct interactions with ICα2, resulting in channel opening or closing. Compared with the corresponding structures of mTMEM16A, TM4–TM7 of AtOSCA1.2 may be more flexible and prone to conformational changes. Upon activation, these helices should undergo tilting and reorganization to open the conduction pore (Fig. 5b). Notably, a long helix parallel to the membrane is also observed in the human mechanically activated ion channel Piezo1 and is believed to act as a beam that transmits membrane tension-induced changes[47–49]. Recently, the atomic structures of AtOSCA1.1, AtOSCA1.2 and AtOSCA3.1 have also been reported by two other groups[50,51]. The overall structures for these members are quite similar with our structure of AtOSCA1.2. Structure alignments between AtOSCA1.1 and our AtOSCA1.2 structure reveal a root mean square deviation of 1.6 Å. Differences mainly lie in the pore helices like TM4–TM7 and the two long cytosolic helices (Supplementary Fig. 9). Meanwhile, they are also quite different from AtOSCA2.2 in the two long helices and the anchor region of the cytosolic domain. Notably, electrophysiological examinations of different OSCA family members indicate quite distinct ion conductance[52]. The cytosolic domain may partly contribute to these different channel properties.

Except directly sensing the mechanical changes in the lipid bilayer, certain osmosensitive channels can also be activated by

changes in the ionic strength as water fluxes, like the volume-regulated anion channel LRRC8[53,54]. It remains elusive whether AtOSCA1.2 can be opened by ion concentration alterations. Besides, whether or not AtOSCA1.2 carries a lipid scrambling function as some TMEM16 family members do is yet to be determined. Without ruling out these possibilities, the detailed function and mechanism for AtOSCA1.2 activation remain to be addressed. Nonetheless, considering the few structural information of osmosensitive channels, the atomic structure of AtOSCA1.2 and the key elements identified in this study provide perspectives for the mechanistic elucidations.

## Methods

**Protein expression and purification**. The cDNA of full-length OSCA1.2 and OSCA2.2 in *A. thaliana* was subcloned into the pFastBac1 vector (Invitrogen) with a carboxyl-terminal $His_{10}$ tag. Primers used for the subcloning can be found in Supplementary Table 1. The recombinant AtOSCA1.2/AtOSCA2.2 was expressed using the baculovirus system (Invitrogen). Briefly, bacmids were generated in DH10Bac cells (Invitrogen). The baculoviruses were generated and amplified in Sf-9 insect cells (Invitrogen). Forty-eight hours after viral infection, cells were collected and resuspended in buffer containing 25 mM HEPES pH 7.4, 150 mM NaCl and 1% (w/v) digitonin, then incubated at 4 °C for 2 h. The insoluble fraction was precipitated by ultracentrifugation at $150,000 \times g$ for 30 min The supernatant was incubated with the Ni-NTA resin (Qiagen) at 4 °C for 30 min The resin was then rinsed three times with wash buffer containing 25 mM HEPES pH 7.4, 150 mM NaCl, 25 mM imidazole and 0.1% digitonin (w/v). The protein was eluted with wash buffer plus 300 mM imidazole. The eluent was concentrated and then subjected to size-exclusion chromatography using a Superose 6 column (GE Healthcare) in buffer containing 25 mM HEPES pH 7.4, 150 mM NaCl and 0.1% digitonin. The peak fractions were pooled together and further concentrated to approximately 5 mg ml$^{-1}$ for EM analysis.

**Sample preparation and cryo-EM data acquisition**. Four-microlitre aliquots of purified AtOSCA1.2 were placed on glow-discharged holey carbon grids (Quantifoil Cu R1.2/1.3, 300 mesh). The grids were blotted for 4 s at 8 °C and 100% humidity and then flash-frozen in liquid ethane using Vitrobot Mark IV (FEI). The grids were then transferred to a Titan Krios (FEI) electron microscope operating at 300 kV with a nominal magnification of ×22,500. Images were recorded manually using the UCSFImage4 software[55] under a K2 Summit electron-counting direct detection camera (Gatan) in super-resolution mode. A total of 5414 images were collected with defocus values varying from −1.6 to −2.5 μm. Each image was acquired with an exposure time of 8 s and dose-fractionated to 32 frames at a total dose rate of 50 $e^-$ Å$^{-2}$ for each stack. The stacks were first motion corrected with MotionCorr[56] and then binned two-fold to a pixel size of 1.307 Å. The output stacks from MotionCorr were further corrected with MotionCor2[57], and dose weighting was performed at the same time[58]. Defocus values were estimated with Gctf[59].

**Image processing**. A simplified flowchart for the image processing procedure is presented in Supplementary Fig. 3a. A total of 4783 micrograph stacks were manually picked for further data processing, and a total of 1,331,485 particles were automatically picked using RELION 2.0[60]. After two-dimensional (2D) classification, 702,922 particles were selected and subjected to a global angular search three-dimensional (3D) classification with 1 class and 50 iterations. The initial model was generated with images of selected 2D class averages using RELION2.0. The results of the last 5 iterations were subjected to a local angular search 3D classification with 3 classes and an angular step of 3.75°. A total of 544,119 good particles were selected from the local angular search 3D classification and merged together. These particles were then subjected to a local angular search 3D auto-refinement, resulting in a 3D reconstruction map with a resolution of 4.08 Å after post-processing. A guided multi-reference 3D classification procedure was then applied to the merged data set using RELION2.0. Particles of the best-classified class were subjected to 3D auto-refinement, resulting in a 3D reconstruction map with a resolution of 3.84 Å after postprocessing. A soft overall mask was generated from one of the two unfiltered half-reconstruction map with an suitable initial binarization threshold using RELION2.0. The map quality was improved when the particles were subjected to 3D auto-refinement with the soft overall mask applied, and the resolution of the reconstruction map after postprocessing reached 3.68 Å. The resolution was estimated with the gold-standard FSC 0.143 criterion[61] with a high-resolution noise substitution method[62]. Local resolution variations were estimated using ResMap[63].

**Model building and refinement**. The 3.68 Å reconstruction map of AtOSCA1.2 sharpened with an automatically estimated *B*-factor of −224 Å$^2$ was used for de novo model building in COOT[64]. Bulky residues such as Phe, Tyr, Trp and Arg were used to guide the sequence assignment, and the chemical properties

of amino acids were considered to facilitate model building. Structure refinements were carried out by PHENIX in real space using phenix.real_space_refine with secondary structure and geometry restraints to prevent structure overfitting[65]. Overfitting of the model was monitored by refining the model in one of the two independent maps from the gold-standard refinement approach and testing the refined model against the other map (Supplementary Fig. 2g). Statistics on the 3D reconstruction and model refinement can be found in Supplementary Table 2.

## Data availability

Data supporting the findings of this manuscript are available from the corresponding authors upon reasonable request. The 3D cryo-EM density map of AtOSCA1.2 in digitonin and AtOSCA2.2 have been deposited in the Electron Microscopy Data Bank under the accession number EMD-9682 and EMD-9677, respectively. Coordinates for the AtOSCA1.2 structure model have been deposited in the PDB under the accession code 6IJZ. A reporting summary for this article is available as a Supplementary Information File. The source data underlying Supplementary Fig. 2a is provided as a Source Data file.

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

## Acknowledgements

We thank Q. Zhou, C. Yan, and H. Qian for discussions of data processing and structure determination; J. Lei and X. Li for technical support on cryo-EM data collection of AtOSCA1.2; Z. Guo and X. Huang for technical support on cryo-EM data collection of AtOSCA2.2; and the Tsinghua University Branch of the China National Center for Protein Sciences (Beijing) and the Center for Biological Imaging (CBI), Institute of Biophysics, Chinese Academy of Science for the EM facility support. This work was supported by funds from the University of Science and Technology of China (KY2070000059 to L.S.) and the Chinese Ministry of Science and Technology (2015CB910104 to J.W.). L.S. is supported by an Outstanding Young Scholar Award from the Qiu Shi Science and Technologies Foundation.

## Author contributions

L.S. conceived the project. All the authors designed the experiments. X.L. and L.S. performed most of the experiments. J.W. conducted the data acquisition and model building. All authors contributed to the data analysis and manuscript preparation. X.L. and L.S. wrote the paper.

## Additional information

**Competing interests:** The authors declare no competing interests.

