## [Peer Review File · Nature Communications]

Reviewers' Comments:

Reviewer #1:

Remarks to the Author:

In this manuscript, Liu et al. presented an *Arabidopsis thaliana* mechanosensitive channel structure from the OSCA family of proteins. While presented cryo-EM structural work is good, this manuscript needs improvements before it could be published.

Comments:

- 1) Authors determined a structure of the OSCA 1.2 protein according to the OSCA family of proteins nomenclature. It is not clear to me why they prefer to call this protein CSC1. Please explain this clearly or remove this from the text as it is very confusing.
- 2) Authors should re-write their Introduction section of the paper. It is very poorly written and does not even mention Piezo channels that have been clearly shown to play a major role in mechanosensation.
- 3) Authors identified RNA recognition motif in the structure however it is not clear if this protein binds RNA. Can authors explain more about the significance of this motif for protein function and gating.
- 4) Structure of the OSCA2.2 is at 5.4Å, nevertheless this information gives authors a unique opportunity to describe the differences between OSCA family members. It is not clear to me, why they put this information into the Supplementary Figure 8. I think it should be moved in the main text of the manuscript.
- 5) Supplementary Figure 9 also should be in the main text as it is a conclusion figure with the proposed mechanism of gating.

Reviewer #2:

Remarks to the Author:

Sun, Wang and colleagues present a cryo-EM structure of hyperpolarization-gated calcium permeable channel CSC1. The structure that is presented is interesting and could have potential to provide novel insight. However, the manuscript in its current form is not suitable for publication in Nature Communications without addressing issues below.

Major comments

1. The statistics for the final model indicates that the model was below the acceptable level. Based on the Ramachandra, only 87 % lies in the favorable region. Structures at the similar resolutions (3.7 Å) typically show > 95% favorable region in their Ramachandra plot. I also realized that the authors had to apply a very high B-factor (-224 Å²) to the data for model building, which is surprising given the resolution of the reconstruction. The authors need to improve their model by re-refining the model. The authors need to do the followings.

- The authors need to show the quality of the map at a milder B-factor sharpening (-100 Å²).
- Clash score (preferably below 10)
- Molprobit score (below 3.5)
- Rotamer outlier
- Provide FSC value of model-map based on Phenix RSR.
- Description of the model refinement in the method
- Describe how the mask that was used for the final reconstruction was created in the methods.

This mask should be included in a reconstruction flow chart as well.

2. Osmosensitivity can be generated by either mechanoinsensitivity or salt sensitivity. For example, the Dutzler group showed the volume regulated LRRC opens by salt sensing, not by mechanosensing. Therefore, it is possible that CSC1 is also a salt sensing ion channel rather than a mechanosensitive ion channel. However, the authors did not explain this possibility and defines CSC1 as a mechanosensitive ion channel. Authors must discuss the possibility of CSC1 not being mechanosensitive.

3. The dimerization being limited to the cytoplasmic domain is unique. Please provide the area for the dimer interaction.

Reviewer #3:

Remarks to the Author:

The manuscript by Liu et al provides the first structure of CSC1, an osmosensitive plant channel. It falls within the class of Ca-CIC channels which are poorly understood both functionally and structurally. The authors make useful comparison of their new structure and related channels. The structural aspects of this manuscript are of high quality. However, the paper gets into trouble on the functional aspects of this putative channel. In figure 5 they show an increase in the outwardly rectifying current in response to sorbitol application. Presumably, this raises the osmolarity of the external saline where the cell would shrink as water move out. So this is opposite of swell-type current, commonly reported for other members like TMEM16A. Yet the current is not defined, is it outwardly running cations (Na⁺ or Ca²⁺) or inwardly running Cl⁻? The authors should keep in mind that there are unsettled controversies regarding the related TMEM16 family members function (channel, lipid scramblase or both)—might CSC1 also carry dual functions? Furthermore there are reports of osmotically activated, outwardly rectify, anion and cationic conductance's in HEK cells –indeed this is how they survive osmotic stress. The current magnitude does not appear to be of sufficient size to be attributed to an overexpressed channel, but it is impossible to tell because the data should be reported as average current density which accounts for the size of the cell. The record in Supplementary Figure 7 appears to be an artifact of swelling the cell and is not convincing. Undoubtedly the seal resistance is lost due changing the membrane turgor, resulting in unstable current. The functional analysis presented in this manuscript is exceedingly superficial. If the authors can't improve upon them in resubmission, then we suggest removing it entirely. In which case, we would STILL support its publication.

There are grammatical errors and vague statements throughout the manuscript– some of which we identify below→. These should be fixed by the authors.

- 1) Abstract "Cells respond to osmotic stress via a complex signaling network to make adjustments at various levels." This sentence is so vague that it is meaningless.
- 2) "In eukaryotes, calcium ion acts as a primary regulator of the initial responses to osmotic pressure⁵⁻⁷. The first event observed after osmotic stress treatment is a rapid increase in the cytosolic free Ca²⁺ concentration^{5,8,9}." As written, the sentence suggests that this shared by all eukaryotes, but then the authors only site works done in plants.
- 3) "While the detailed mechanism for AtCSC1 activation remains to be addressed, considering the few structural information of mechanosensitive channels, the atomic structure of AtCSC1 and the key elements identified in this study provide new perspectives for the mechanistic elucidations, especially in economic and food crops". Grammar etc. Economics appears to be an unnecessary extrapolation of the scope and impact.

Overall Response to Reviewers' Comments:

We thank all three reviewers for their constructive and helpful comments. Before we address the specific comments from each reviewer, we would like to summarize the major changes in the revised manuscript:

1. We have polished the writing of the introduction and discussion parts of the manuscript according to the reviewers' suggestions. We have corrected the grammar errors and vague statements, made a clearer and more comprehensive discussion about the function and activation mechanism of AtCSC1 .
2. We have further refined our structure model and significantly improved its quality for publication. In the Ramachandran plot, 88.77% of the residues lie in the preferred region, 10.93% lie in allowed region and 0.3% are outliers. The all-atom clash score after refinement is 4.03. And the Molprobity score is 1.77.
3. To emphasize the structural part of our study and ease the reviewer's concerns about the functional analysis, we have moved these results to the supplementary materials (as Supplementary figures 7 and 8 in the revised manuscript). The previous supplementary figure about the EM map of AtOSCA2.2 and the comparison with AtCSC1 is now moved into the main text, along with the proposed model for AtCSC1 activation (as Figure 5 in the revised manuscript).
4. In the revised manuscript, we have discussed more about the unique dimerization pattern of AtCSC1 and the role that RRM motif plays in the dimer formation (page 12). A panel is also added in Supplementary Figure 4d to show the dimer interface.

Reviewer #1:

We thank this reviewer for his/her constructive and helpful comments. S/he raised five specific points that are addressed below:

- 1) *Authors determined a structure of the OSCA 1.2 protein according to the OSCA family of proteins nomenclature. It is not clear to me why they prefer to call this protein CSC1. Please explain this clearly or remove this from the text as it is very confusing.*

This protein was originally named as At4G22120 after the gene name that was identified in *Arabidopsis*. In 2014, it was for the first time identified as an ion channel that could be activated by hyperosmotic shock through large genetic screens by Hou *et al.* (*Cell Research*. 2014 May; 24:632-635.). Thus, this group renamed it AtCSC1, short for *Arabidopsis* Calcium permeable Stress-gated cation Channel 1. Later in 2014, another group led by Zhen-ming Pei identified an *Arabidopsis* mutant that exhibit low hyperosmolality induced $[Ca^{2+}]_i$ increase and named the mutant as *osca1* (*Nature*. 2014 Oct 14; 514(7522): 367-371.). Eight OSCA1 homologues were identified in *Arabidopsis*, including the previously identified AtCSC1 which was named AtOSCA1.2 as to the OSCA nomenclature. In this respect, these two groups independently identified the physiological function of this protein family and give this protein two distinct names that are both used in later literatures. We chose to use AtCSC1 in our manuscript mainly because of the time this protein's function was first identified.

Besides, in the Transporter Classification Database (TCDB), which is a comprehensive classification system for membrane transport proteins, a specific entry (#1.A.17.5) containing AtCSC1 and its homologues in fungi and human was named the Calcium-permeable Stress-gated Cation Channel (CSC) Family. Thus, CSC1 may be a more representative name in different species.

Concerning the time this protein's function was first identified and the representative entry name in TCDB, we chose to use AtCSC1 in the manuscript, with a noted alias AtOSCA1.2. To avoid the confusion that might brought as this reviewer suggested, we have revised the introduction part of the manuscript and explained it more clearly.

- 2) *Authors should re-write their Introduction section of the paper. It is very poorly written and does not even mention Piezo channels that have been clearly shown to play a major role in mechanosensation.*

Point accepted. We have revised the introduction section of the manuscript according to the reviewer's suggestion. The logic should be more clear after revision and the mammalian mechanosensitive channels including the potassium channels TREK-1 and TRAAK, TRP family channels like TRPV4 and TRPC6, and the Piezo channels are now included in this section. Besides, we have also discussed the structural element of Piezo1 in the Discussion section of the manuscript (page 15):

“Notably, a long helix parallel to the membrane is also observed in the human mechanically activated ion channel Piezo1 and is believed to act as a beam that transmits membrane tension-induced changes⁴⁷⁻⁴⁹.”

- 3) *Authors identified RNA recognition motif in the structure however it is not clear if this protein binds RNA. Can authors explain more about the significance of this motif for protein function and gating.*

We have considered the possibility that this motif may bind RNA and function in RNA regulation. However, the classical β -sheet surface for RNA binding is occupied in the current conformation by the loop prior to TM4 and the carboxyl-terminus of AtCSC1 (as shown in Supplementary Fig. 9b). Unless there is a large conformation change of this part, AtCSC1 is unlikely to bind RNA. As we have discussed in the manuscript, the RNA recognition motif seems to play a major function in mediating the dimeric architecture of AtCSC1. Unlike nhTMEM16 and mTMEM16A, in which the dimer formation is mediated by direct interaction between the transmembrane helices, AtCSC1's dimer formation is solely mediated by its cytosolic domain, which mainly consists of the RRM motif and the short helix after TM11.

The dimer interface is formed by both hydrophobic interactions mediated by W331, V335 and L685, and hydrogen bonds formed by Q340 and R682, as shown below. We have tried to break the hydrogen bonds by single or double mutating the residues to alanines in order to break the dimer formation of AtCSC1 and analyze the function of the monomeric AtCSC1. However, we have not succeed with this strategy and the Q340A and R682A double mutant yield hardly any protein expression. This also suggests an essential structural role for the RRM motif and the dimeric interface.

Besides, the RRM motif identified in AtCSC1 differs from the canonical one with the addition of the two unique long helices, IC α 2 and IC α 3 (as shown in the figure below). As the electrophysiological analysis result indicates, these two long helices and the linker in between are import to channel activation under hyperosmotic stimuli. This might be the result of protein evolution to adopt such a structural element to sense the osmotic stress.

In the revised manuscript, we discussed more about the dimer interface (page 12) and also added a panel to clearly show the area of dimer interface in Supplementary Figure 4d.

- 4) *Structure of the OSCA2.2 is at 5.4A, nevertheless this information gives authors a unique opportunity to describe the differences between OSCA family members. It is not clear to me, why they put this information into the Supplementary Figure 8. I think it should be moved in the main text of the manuscript.*

Point accepted. We have moved the Supplementary Figure 8 into the main text as Figure 5a.

- 5) *Supplementary Figure 9 also should be in the main text as it is a conclusion figure with the proposed mechanism of gating.*

Point accepted. We have moved the Supplementary Figure 9 into the main text as Figure 5b.

Reviewer #2:

We thank this reviewer for his/her constructive and helpful comments. S/he raised three specific points that are addressed below:

- 1) *The statistics for the final model indicates that the model was below the acceptable level. Based on the Ramachandra, only 87 % lies in the favorable region. Structures at the similar resolutions (3.7 Å) typically show > 95% favorable region in their Ramachandra plot. I also realized that the authors had to apply a very high B-factor (-224 Å²) to the data for model building, which is surprising given the resolution of the reconstruction. The authors need to improve their model by re-refining the model. The authors need to do the followings.*
 - *The authors need to show the quality of the map at a milder B-factor sharpening (-100 Å²).*
 - *Clash score (preferably below 10)*
 - *Molprobability score (below 3.5)*
 - *Rotamer outlier*
 - *Provide FSC value of model-map based on Phenix RSR.*
 - *Description of the model refinement in the method*
 - *Describe how the mask that was used for the final reconstruction was created in the methods. This mask should be included in a reconstruction flow chart as well.*

We thank this reviewer for the helpful comments.

a. As to the B-factor applied during postprocessing, we choose to estimate the B-factor automatically in Relion, which use a procedure described by Rosenthal and Henderson (2003, JMB), and the resulted B-factor value is -224 Å². The applied B factor is validated by visual judgment that the side chain densities for most of the residues are distinguishable. Meanwhile, we have also tested a lower B factor such as -150 Å² and -100 Å² as shown below:

As viewed from the maps with different B-factor applied, the one with a B-factor of -224 Å² is more suitable for residue assignment and model building. And at a B-factor of -100 Å², we can hardly assign side chains. We can also tell this from a zoom-in view of the density map for the transmembrane segments, such as TM6 as shown below:

Thus, we chose to use the map generated with a B-factor estimated automatically of -224 \AA^2 for the model building. We also noticed that by applying the energy filter during EM image collection would greatly lower the applied B-factor. A good example is for the structure determination of a eukaryotic sodium channel Nav_vPaS . Without using energy filter, the B-factor applied to generate the density map at 3.8 \AA is -270 \AA^2 (*Science*. 2017 Mar 03; 355(6328), which is even higher than that we used. With energy filter, the B-factor is only -100 \AA^2 (*Science*. 2018 Jul 26: eaau2596.).

With this, we believe the B-factor we applied is a reasonable value to this data set, which helps the model building.

b. As this reviewer suggested, we have further refined our model and significantly improved the quality of the structure. The FSC curve between model and map is provided in Figure S2g. The all-atom clash score after refinement is 4.03. And the Molprobtity score is 1.77. In the Ramachandran plot, 88.77% of the residues lie in the preferred region, 10.93% lie in allowed region and 0.3% are outliers. The rotamer outlier is only 0.52%. The best we can get based on the current resolution and EM density map is to make the favorable region 88.77% in the Ramachandra plot. In fact, as we looked through the EM Data Bank, many structures determined by single particle cryoEM analysis at resolutions around 3.7 \AA or even higher cannot reach such a high standard (>95%). Listed below are some examples:

Protein	Resolution (Å)	PDB-ID	Favoured	Allowed	Outliers	Reference
TMEM16A	3.8	6BGI	84%	16%	0%	Nature (2017), 552: 426-429.
NavPaS	3.8	5X0M	82%	13%	5%	Science (2017), 355(6328).
MBH	3.7	6CFW	89%	11%	0%	Cell (2018), 173: 1636.
Patched1	3.5	6D4H	90%	9%	0%	Nature (2018), 560:

TRPM7	3.7	6BWD	86%	12%	1%
-------	-----	------	-----	-----	----

Therefore, we believe that our structure model after further refinement is now of enough quality for publication.

c. A panel of the FSC value of model-map has been provided in Supplementary Figure 2g.

d. As this reviewer suggested, we have further described the model refinement procedure in the method section (page 27 of the revised manuscript).

e. As this reviewer suggested, we have described how the mask that was used for the final reconstruction was created in the method section as below (page 27 of the revised manuscript):

“A soft overall mask was generated from one of the two unfiltered half-reconstruction map with an suitable initial binarisation threshold using RELION2.0.”

We have also added the mask in the reconstruction flow chart (as in Supplementary Figure 3a of the revised manuscript).

- 2) *Osmosensitivity can be generated by either mechaosensitivity or salt sensitivity. For example, the Dutzler group showed the volume regulated LRRC opens by salt sensing, not by mechanosensing. Therefore, it is possible that CSC1 is also a salt sensing ion channel rather than a mechanosensitive ion channel. However, the authors did not explain this possibility and defines CSC1 as a mechanosensitive ion channel. Authors must discuss the possibility of CSC1 not being mechanosensitive.*

Point accepted. We thank this reviewer for this comprehensive thinking that we have neglected. As this reviewer stated, the volume-regulated anion channel LRRC8 has been shown to be activated by low intracellular ionic strength instead of mechanical changes in the lipid bilayer by the Patapoutian group (*Cell*. 2016 Jan 28; 164(3): 499–511.) and Dutzler group (*Nature*. 2016 Jun 14; 558(7709): 254-259.). We cannot rule the the possibility of CSC1 being a salt sensing ion channel based on current results. Thereby, as this reviewer suggested, we discussed this possibility as below (page 16 in the revised manuscript):

“Except directly sensing the mechanical changes in the lipid bilayer, certain osmosensitive channels can also be activated by changes in the ionic

strength as water fluxes, like the volume-regulated anion channel LRRC8^{50,51}. It remains elusive if AtCSC1 can be opened by ion concentration alterations.”

- 3) The dimerization being limited to the cytoplasmic domain is unique. Please provide the area for the dimer interaction.

Point accepted. As this reviewer suggested, we have discussed this part more thoroughly in the revised manuscript (page 12):

“As seen in the structure, two α -helices of the RRM ($IC\alpha 1$ and $IC\alpha 4$) together with the short helix following TM11 ($IC\alpha 5$) play mainly a structural role in mediating the dimer formation of AtCSC1 (Supplementary Fig. 4c). Intense hydrophobic interactions are formed by W331, V335 and L685. A pair of hydrogen bond is also identified between Q340 and R682 (Supplementary Fig. 4d).”

Besides, we also added a panel to clearly show the area of dimer interface in Supplementary Figure S4d as shown below:

Reviewer #3:

We thank this reviewer for his/her constructive comments. S/he raised one major concern and three specific points.

There is one major concern about the functional analysis part of CSC1:

“The manuscript by Liu et al provides the first structure of CSC1, an osmosensitive plant channel. It falls within the class of Ca-ClC channels which are poorly understood both functionally and structurally. The authors make useful comparison of their new structure and related channels. The structural aspects of this manuscript are of high quality. However, the paper gets into trouble on the functional aspects of this putative channel. In figure 5 they show an increase in the outwardly rectifying current in response to sorbitol application. Presumably, this raises the osmolarity of the external saline where the cell would shrink as water move out. So this is opposite of swell-type current, commonly reported for other members like TMEM16A. Yet the current is not defined, is it outwardly running cations (Na⁺ or Ca²⁺) or inwardly running Cl⁻? The authors should keep in mind that there are unsettled controversies regarding the related TMEM16 family members function (channel, lipid scramblase or both)—might CSC1 also carry dual functions? Furthermore there are reports of osmotically activated, outwardly rectify, anion and cationic conductance’s in HEK cells –indeed this is how they survive osmotic stress. The current magnitude does not appear to be of sufficient size to be attributed to an overexpressed channel, but it is impossible to tell because the data should be reported as average current density which accounts for the size of the cell. The record in Supplementary Figure 7 appears to be an artifact of swelling the cell and is not convincing. Undoubtedly the seal resistance is lost due changing the membrane turgor, resulting in unstable current. The functional analysis presented in this manuscript is exceedingly superficial. If the authors can’t improve upon them in resubmission, then we suggest removing it entirely. In which case, we would STILL support its publication.”

We really appreciate this reviewer’s constructive comments. As shown by Hou et al. (*Cell Research*. 2014 May; 24:632-635.) and Yuan et al. (*Nature*. 2014 Oct 14; 514(7522): 367-371.), AtCSC1 and AtOSCA1 are cation channels permeable to both monovalent and divalent cations including Ca²⁺, K⁺ and Na⁺. The outward-rectifying current observed in our analysis would be a result of the outwardly running cations. We also noticed that by using different cell systems, different types of current were observed. In the *Xenopus oocytes* system, cells expressing AtCSC1 showed an inward current when treated with hyperosmotic shock. While in the HEK293 cell system, cells expressing AtOSCA1 showed an weak outward-rectifying current. In our manuscript, we used the HEK293 cell system to analyze the function of AtCSC1 and also observed the outward-rectifying current under hyperosmotic stimuli. Cells transfected with the wild type AtCSC1, empty vector or the AtCSC1 mutants

we generated, did show different behavior under the 300 mM sorbitol treatment. These are quite consistent with the results reported by Yuan et al. (as shown below, part of Figure 4 in *Nature*. 2014 Oct 14; 514(7522): 367-371.). We believe that these results show whether the heterologously expressed AtCSC1 is activated or not under the hyperosmotic shock and supports the model we have proposed in the manuscript that the hydrophobic linker and the two cytosolic helices are important to AtCSC1 activation.

Actually, as this reviewer suggested, we have tried to optimize the electrophysiological analysis procedure, to see if we could increase the observed current magnitudes by using higher concentrations of sorbitol treatment, like 1 M sorbitol. However, the observed currents became quite unstable and noisy, indicating that this concentration was too harsh to the cell and could not reflect if AtCSC1 plays a role in the osmotic response.

Indeed, as this reviewer commented, there are unsettled controversies about the TMEM16 family protein functions. As lacking a model system that can restore features of the plant cell, we cannot exclude the possibility that AtCSC1 also carries a lipid scrambling function. Thus, we revised our manuscript by discussing this possibility (page 16).

To emphasize the structural part of our study and further ease the concerns of this reviewer, we also moved the functional analysis results to the supplementary materials (as Supplementary figures 7 and 8 in the revised manuscript).

This reviewer raised four specific points that are addressed below:

- 1) Abstract *“Cells respond to osmotic stress via a complex signaling network to make adjustments at various levels.” This sentence is so vague that it is meaningless.*

Point accepted. We have deleted this sentence and modified the abstract as below (page 2 of the revised manuscript):

“In plants, hyperosmolality stimuli triggers opening of the osmosensitive channels, leading to a rapid downstream signaling cascade initiated by cytosolic calcium concentration elevation.”

- 2) *“In eukaryotes, calcium ion acts as a primary regulator of the initial responses to osmotic pressure⁵⁻⁷. The first event observed after osmotic stress treatment is a rapid increase in the cytosolic free Ca²⁺ concentration^{5,8,9}.” As written, the sentence suggests that this shared by all eukaryotes, but then the authors only site works done in plants.*

Point accepted. We have modified the sentence as below and added three more references about the work done in mammals (page 3 of the revised manuscript):

“In plants and mammals, calcium ion acts as a primary regulator of the initial responses to osmotic pressure¹¹⁻¹⁶. The first event observed after osmotic stress treatment is a rapid increase in the cytosolic free Ca²⁺ concentration^{17,18}.”

- 3) *“While the detailed mechanism for AtCSC1 activation remains to be addressed, considering the few structural information of mechanosensitive channels, the atomic structure of AtCSC1 and the key elements identified in this study provide new perspectives for the mechanistic elucidations, especially in economic and food crops”. Grammar etc. Economics appears to be an unnecessary extrapolation of the scope and impact.*

Point accepted. We have revised this sentence as below (page 16 of the revised manuscript):

“Nonetheless, considering the few structural information of osmosensitive channels, the atomic structure of AtCSC1 and the key elements identified in this study provide new perspectives for the mechanistic elucidations.”

Reviewers' comments:

Reviewer #1 (Remarks to the Author):

Authors addressed all my concerns. I recommend this manuscript for publication.

Reviewer #2 (Remarks to the Author):

These authors have carefully addressed the most concerns of this reviewer. Although these authors' model building is not optimal in comparison with two other structures with similar resolution that were published very recently, the overall structural quality is adequate. Note that, Zhang et al. published the structures of OSCA1.1 and OSCA3.1 in Nature Structural & Molecular Biology on Sep. 6, 2018. Jojoa-Cruz et al. published the structures of OSCA1.2 in BioRxiv on Sep. 4, 2018. At the same time from the same group, Murthy et al. published a paper entitled "OSCA/TMEM63 Are an Evolutionarily Conserved Family of Mechanically Activated Ion channels" in BioRxiv on Sep. 4, 2018, in which the reconstitution and/or functional expression of OSCA1.2, OSCA1.8, OSCA2.3, OSCA3.1 and OSCA4.1 were carried out. These papers should be cited appropriately in the manuscript and clear comparisons should be made. In addition, for the comparison of these studies and structures, the name of "CSC1" in the title and throughout the manuscript should be revised to "OSCA1.2" before it is accepted for publication.

Reviewer #3 (Remarks to the Author):

The authors have answered the MINOR points listed but ignored two MAJOR points of revision. To be clearer, I will restate/paraphrase and list the major points that the authors should correct, all of which relate the superficial analysis of the CSC1's putative ion channel function.

1) The authors assert (and re-assert in the rebuttal) that these are cationic currents, yet they offer zero evidence. They do however cite other work, nonetheless, the work they cite also fails to convince me of the ionic identity of the putative currents. They need to try different cation conditions, with and without chloride and determine that the conductance is dependent on cations.

2) Differences in magnitudes should be expressed as current density (pA/pF) instead of current alone. (Sup FIG 7 and 8) This doesn't account for cell size. Given that their overexpression doesn't produce large currents-- I can easily get nA of current from a voltage ramp under these conditions. 1) While patching an HEK cell. This can occur in one of two ways 1) high leak brought on by osmotic shrinking (high external sorb) or 2) I simply patch a large HEK cell.

My position remains the same, regarding the electrophysiology-- If the authors can't improve upon this aspect in resubmission, then we suggest removing it entirely. In which case, we would STILL support its publication.

Reviewer #2:

These authors have carefully addressed the most concerns of this reviewer. Although these authors' model building is not optimal in comparison with two other structures with similar resolution that were published very recently, the overall structural quality is adequate. Note that, Zhang et al. published the structures of OSCA1.1 and OSCA3.1 in Nature Structural & Molecular Biology on Sep. 6, 2018. Jojoa-Cruz et al. published the structures of OSCA1.2 in BioRxiv on Sep. 4, 2018. At the same time from the same group, Murthy et al. published a paper entitled "OSCA/TMEM63 Are an Evolutionarily Conserved Family of Mechanically Activated Ion channels" in BioRxiv on Sep. 4, 2018, in which the reconstitution and/or functional expression of OSCA1.2, OSCA1.8, OSCA2.3, OSCA3.1 and OSCA4.1 were carried out. These papers should be cited appropriately in the manuscript and clear comparisons should be made. In addition, for the comparison of these studies and structures, the name of "CSC1" in the title and throughout the manuscript should be revised to "OSCA1.2" before it is accepted for publication.

We thank this reviewer for his/her helpful comments. As s/he suggested, we have cited the recent publications mentioned above in our revised manuscript (page 15, reference number 50-52).

We have compared our AtOSCA1.2 and AtOSCA2.2 structures with the released AtOSCA1.1 and AtOSCA3.1 structures determined by Zhang et al. The overall structure of AtOSCA1.2 and AtOSCA1.1 are quite similar, with a root mean square deviation of 1.6 Å. Differences mainly lie in the pore-forming helices like TM4-7 and the two long cytosolic helices. A supplementary figure is now added in the revised manuscript (Supplementary Figure 9) to show the structure alignments results. Meanwhile, the AtOSCA2.2 structure we have determined is still quite unique, which is lack of the two long helices and the anchor region in the cytosolic domain. This may in part contribute to the different channel properties revealed Murthy et al. Besides, it also makes the AtOSCA2.2 structure more meaningful to the study of OSCA family channels.

As this reviewer suggested, we have changed the name of CSC1 to OSCA1.2 in our revised manuscript.

Reviewer #3:

The authors have answered the MINOR points listed but ignored two MAJOR points of revision. To be clearer, I will restate/paraphrase and list the major points that the authors should correct, all of which relate the superficial analysis of the CSC1 's putative ion channel function.

1) The authors assert (and re-assert in the rebuttal) that these are cationic currents, yet they offer zero evidence. They do however cite other work, nonetheless, the work they cite also fail to convince me of the ionic identity of the putative currents. They need to try different cation conditions, with and without chloride and determine that the conductance is dependent on cations.

2) Differences in magnitudes should be expressed current density (pA/pF) instead of current alone. (Sup FIG 7 and 8) This doesn't account for cell size. Give that their overexpression doesn't produce large currents-- I can easily get nA of current from voltage ramp under these conditions. 1) While patching an HEK cell. This can occur in one of two ways 1) high leak brought on by osmotic shrinking (high external sorb) or 2) I simply patch a large HEK cell.

My position remains the same, regarding the electrophysiology-- If the authors can't improve upon this aspect in resubmission, then we suggest removing it entirely. In which case, we would STILL support its publication.

We sincerely thank and respect this reviewer for his/her constructive comments. We have realized the imperfectness of our electrophysiology assay system and the lack of solid evidence to the protein's function. Since we are not able to improve it in a relatively short time and considering the fierce competition of this project, we removed the electrophysiology results from the manuscript as this reviewer suggested and revised our manuscript accordingly. The revised manuscript now focuses on the structural interpretations of these two proteins and still provide meaningful insights into this kind of membrane proteins. In this way, we truly thank this reviewer for his/her support of its publication.